# Impact and perceptions of Active Learning Classrooms on reducing sedentary behaviour and improving physical and mental health and academic indicators in children and adolescents: A scoping review

**Mairena Sánchez-López**[1,2]*, **Jesús Violero-Mellado**[1,2], **Vicente Martínez-Vizcaíno**[1,3], **Arto Laukkanen**[4], **Arja Sääkslahti**[4], **María Eugenia Visier-Alfonso**[1,5]

**1** Social and Health Research Center, Universidad de Castilla-La Mancha, Cuenca, Spain, **2** Faculty of Education, Universidad de Castilla-La Mancha, Ciudad Real, Spain, **3** Facultad de Ciencias de la Salud, Universidad Autónoma de Chile, Talca, Chile, **4** Faculty of Sport and Health Sciences, University of Jyväskylä, Jyväskylä, Finland, **5** Faculty of Nursing, Universidad de Castilla-La Mancha, Cuenca, Spain

\* mairena.sanchez@uclm.es

## Abstract

Prolonged sitting in school harms children's physical and mental health and reduces the ability to focus on classroom tasks. 'Active Learning Classrooms' (ALCs) aim to decrease sitting time, following current pedagogical trends, though research on the effects of ALCs on these aspects is still an emerging field. The aims of this review were to: (i) synthesise the available literature on the impact of ALCs on reducing sedentary behaviour, increasing physical activity (PA), physical and mental health, and academic indicators in children and adolescents; and (ii) describe the educational community's perceptions and teaching practices used in ALCs. This scoping review followed Joanna Briggs Methods and PRISMA guidelines for scoping reviews. We searched for peer-reviewed quantitative and qualitative studies published in English that examined the impact of ALCs on movement patterns, physical or mental health, and academic indicators in children and adolescents, as well as those that explored the perceptions of members of the educational community and the teaching practices used in ALCs. Databases research included MEDLINE (PubMed), ERIC, SCOPUS and ProQuest Education. Nineteen studies were included, of which 11 were experimental, 4 were cross-sectional, and 4 were qualitative. The analysis revealed a predominantly positive influence of ALCs on children's sedentary behaviour, learning engagement and psychological well-being; and mixed results on PA, physical health and academic performance. Our results also suggest that learning spaces are positively perceived and well accepted by the entire educational community, and that teachers teaching in ALCs are more prone to use student-centered and collaborative pedagogies than in traditional classrooms. Although this review shows a positive impact on key health and education variables, the evidence is limited and lacks depth. In addition, the small number of studies and their methodological weaknesses prevent robust conclusions, but the results still help to guide future decisions.

**Data Availability Statement:** All relevant data can be found in the manuscript and its supporting information files.

**Funding:** This review was conducted while Mairea Sánchez-López was benefiting from a research grant awarded by the Spanish Ministry of Universities (Senior Mobility Grants 'Salvador Madariaga 2022', reference number: PRX22/00245). The founders had no role in study design, data collection and analysis, decision to publish, or preparation of the manuscript.

**Competing interests:** The authors have declared that no competing interests exist.

# Introduction

Given the significant amount of time schoolchildren spend in the classrooms, there is a broad consensus that educational institutions have a responsibility not only to provide quality learning opportunities, but also to ensure that this learning takes place in environments that are conducive to the harmonious physical, social, and emotional development of students. Although children tend to be naturally active, they are constantly exposed to opportunities and environments that encourage sedentary behaviour [1], which is defined as 'any waking behaviour characterised by an energy expenditure $\leq 1.5$ metabolic equivalents while in a sitting, reclining or lying position' [2]. Sedentary behaviour can be assessed using report-based methods, such as questionnaires, and device-based methods, such as accelerometers, which are motion sensors that record signals of the magnitude and frequency of body acceleration and can be used to measure the frequency, intensity and duration of movement [3].

Low levels of physical activity (PA) and sedentary lifestyle in children and adolescents have been associated with deleterious effects on cardiometabolic health, symptoms of depression, decreased self-esteem, and reduced ability to focus on classroom tasks [4–8]. School time is responsible for 65%-80% of the periods of uninterrupted sedentary behaviour [9, 10]. Conversely, recent studies have suggested beneficial effects of PA break on neurocognitive functioning and academic performance [11, 12]. Furthermore, sedentary behaviour in childhood tends to persist into adolescence and adulthood [13].

Recent trends in the design of learning environments are to replace the 'traditional classroom' (rows of fixed desks and chairs in front of the teacher's desk) by the so-called 'active permissive classrooms', 'flexible learning spaces' or 'active learning space or classrooms' (hereafter ALCs) [14]. Although it is challenging to find a universally accepted definition for this term, in this review, ALCs refers to educational environments designed to facilitate not only physical movement within the classroom, but also active student-centred participation, even if unplanned. These spaces typically include flexible, non-traditional furniture arrangements that support various collaborative and individualised working configurations, encouraging students to take an active role in their learning [15]. For example, it has been found that children and adolescents spent less time sitting in classrooms that use flexible spaces, and that they accumulate more sitting breaks than in classrooms that are traditionally furnished and arranged [16–19]. ALCs have also been positively associated with wellbeing and mental health [20, 21], although the evidence is very limited. Finally, traditional classroom design is associated with teacher-centred pedagogical practices. In contrast, ALCs are associated with student-centred pedagogical approaches that encourage interaction, collaboration, and student engagement in learning [22]. However, the effect of learning in an open, unconstrained space, in terms of reducing sedentary time, increasing PA and improving children's cognition, psychological wellbeing and engagement in learning has not yet been well demonstrated in well-designed cluster randomised trials.

Therefore, as teachers adopt active pedagogical practices, concerns about the design and impact of learning spaces are increasing. Several literature reviews have been published on this topic. Kariippanon's review of studies up to 2017 described a reduction in sitting time in class and a positive effect on a range of academic indicators in adolescents engaged in ALCs interventions [23]. Talbert and Mor-Avi's 2019 review found a positive influence of flexible learning spaces on learning outcomes and student engagement, focusing mainly on university students [15]. However, neither review included studies that would show the effect of this type of intervention on variables related to student well-being or mental health, nor did these reviews answer key questions that are necessary to ensure the viability and scalability of such interventions in the future, such as student and teacher perceptions.

To address the growing interest in how classroom design influences student learning and well-being, while also addressing sedentary behaviour in school, and given the emergent and heterogeneous nature of the literature on this topic, we chose to use a scoping review methodology. This approach maps key concepts and available sources of evidence and is appropriate for complex or under-reviewed areas [24, 25].

The objectives of this scoping review were to: (i) synthesise the existing literature on the impact of ALCs on reducing sedentary behaviour, increasing PA, well-being and academic indicators in children and adolescents; and (ii) describe the educational community's perceptions and teaching practices used in ALCs.

## Material and methods

### Protocol and registration

We conducted this scoping review using guidance from the Joanna Briggs Methods Manual for Scoping Reviews [26] and reported according to the PRISMA Extension for Scoping Reviews (PRISMA-ScR) (S1 Table) [27]. The review protocol was registered in the Open Science Framework database (https://osf.io) with registration number: https://doi.org/10.17605/OSF.IO/8N6RY.

### Search strategy

The search was conducted in September-December 2023. This was limited to studies that had been peer-reviewed and published in English. PubMed (Medline), ERIC, SCOPUS, and ProQuest Education were searched from inception to December 2023. Keywords and controlled vocabulary where appropriate were used to describe ALCs, sedentary behaviour, physical and mental health and academic indicators concepts in each database (S2 Table). In addition, a hand search of the reference lists of identified articles was undertaken and Google Scholar was used to identify any other primary sources within grey literature. Two reviewers (MSL and JVM) conducted independently the search in electronic databases.

### Eligibility and exclusion criteria

Studies were included if they met the following criteria:

i.  were conducted in pre-school, primary, or secondary school classrooms (age range: 3–18 years) within standard classroom settings;

ii.  reported on the impact of ALCs in reducing sedentary behaviour and improving the physical and mental health and academic indicators of students, with at least one variable reported in the following categories: movement behaviour (e.g., PA, sedentary time, standing, and sitting time); physical health (e.g., adiposity, pain); mental health (e.g., psychological well-being, stress); or academic indicators (e.g., grades, cognition, engagement). In this review an ALC refers to educational environments designed to facilitate not only physical movement within the classroom, but also active student-centred participation, even if unplanned. These spaces often include flexible and non-traditional furniture arrangements that support a variety of collaborative and individualised work configurations.

iii.  examined the perceptions of students, teachers, families, or staff regarding ALCs, including aspects such as acceptability, perceived barriers and facilitators, or the teaching practices employed in ALCs.

Quantitative, qualitative and mixed methods study design were included.

Studies were excluded if they met any of the following criteria:

i. focused on higher education (e.g., universities) or mixed groups (e.g., school-aged children or adolescents and adults);

ii. involved special population groups (e.g., children with clinically significant behavioural disorders, such as attention difficulties, or children with overweight/obesity);

iii. investigated aspects of the built environment in the classroom, including factors such as temperature, light, color, noise, or overall environmental quality;

iv. included interventions that did not align with the definition of ALCs established in this review, as well as those solely involving the replacement of traditional desks with standing desks, fit balls, or pedal desks. These interventions were excluded because they primarily represent changes in posture during classes and do not necessarily modify classroom configuration or teaching-learning dynamics, even if such changes occur incidentally;

v. full-text access was unavailable, and the authors did not respond to requests for additional data or the complete text;

vi. were categorized as text or opinion articles, conference abstracts, doctoral theses, dissertations, or review articles.

## Study selection

Studies were screened for eligibility by title and abstract independently by two researchers (MSL and JVM). After comparing results, discrepancies were resolved by a third reviewer (MVA). Search results were downloaded into Endnote software (Clarivate Analytics) after removing duplicates. The results of the search and the study inclusion process were presented in a PRISMA-ScR flow diagram [27].

## Extraction and charting of results

We extracted data regarding study characteristics (author/s, year, country, study design, participants, school level, intervention and comparator, aim and outcomes) and main results of the studies. Two reviewers (MSL and JVM) using a standardized excel spreadsheet developed by the reviewers extracted the data from each included study and a third author (MVA) arbitrated unresolved disagreements regarding data extraction. Where necessary, study authors were contacted to request missing or additional data.

## Data analysis and synthesis

A narrative synthesis was conducted to summarize the results, and the main findings are presented in evidence tables. Additionally, we performed a Word Cloud figure to provide a visual representation of text in which words are displayed in different sizes and colours according to their frequency in the dataset. The words that appear in larger sizes and brighter colours represent the terms that are repeated in the studies to define the ALCs.

# Results

The electronic search identified 1352 references. After removal duplicates, 1004 studies were screened, and then 19 eligible publications were included [16–22, 28–39], which represented 19 independent studies. The search results and the selection process of studies are presented in

a flow chart (Fig 1). Studies excluded after reading the full text with reasons for exclusion are shown in supplementary information (S3 Table).

As shown in Table 1, all studies were published between 2004 and 2023 and conducted in Australia (n = 5) [17, 22, 29, 31, 39], Belgium (n = 1) [17], Canada (n = 2) [20, 21], Finland (n = 5) [32–35, 37], Israel (n = 1) [38], New Zealand (n = 2) [19, 30] and USA (n = 3) [16, 28, 36]. The students' sample sizes ranged from 24 to 206 and the mean age ranged from 8 to 17 years. Most were conducted in primary schools (n = 13) [16, 18–21, 28, 30, 32–37], four in

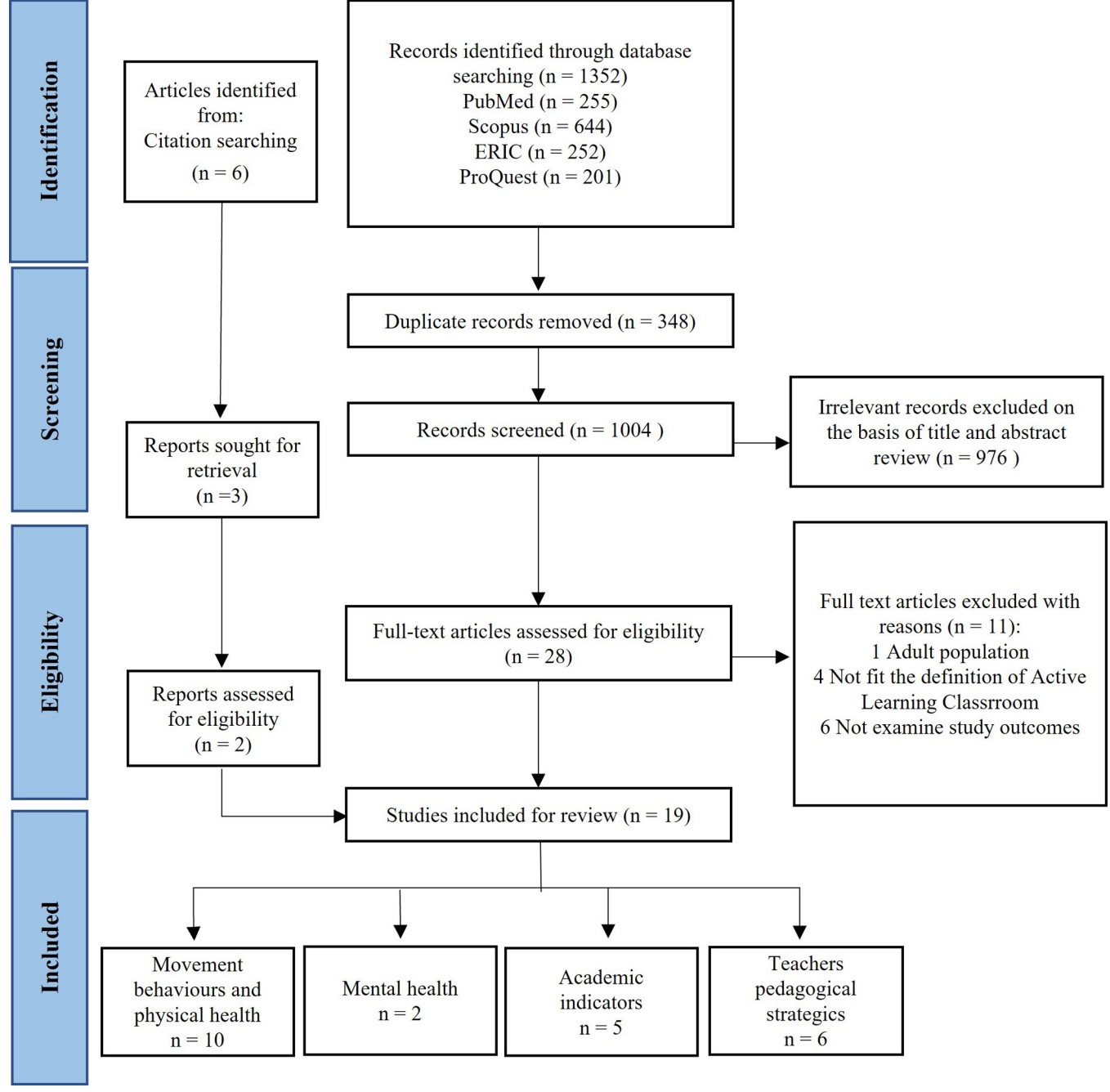

**Fig 1. PRISMA flow chart of the study selection process.**

**Table 1. Characteristics of the studies.**

| Author/s, year | Country | Study design | Participants | School (Primary/ Secondary) | Age (Mean ± SD), or range | Intervention and comparator |
|---|---|---|---|---|---|---|
| Aminian S, et al., 2015 [19] | New Zealand | Mixed method (NRCT + qualitative) | 2 classrooms (n = 1 intervention) of 2 schools, 26 children (n = 18 intervention) Qualitative study: teacher (n = 1) and students (n = 4) | P | 9.8±0.4 Teachers' characteristics: NR | IG: Dynamic classroom design: Traditional desk and chairs were removed from the classroom and replaced with five height-appropriate standing workstations. When children needed to sit, were offered active sitting like beanbags, bench and a "mat space". CG: Traditional desk and chairs. • 9 weeks duration. |
| Attai S, et al., 2019 [28] | USA | NRCT | 10 classrooms (n = 4 intervention) of 1 school, 206 children (n = NR intervention) | P | 8–10 | IG: Flexible classroom furniture defined with the Flexible Environment Learning Scale such as 'High flexibility': students have 4 out of the 4 choices: type of work surface, height of work surface, type of seating, height of seating; all furniture can be moved and most/all are easily transportable by students, and teacher space consumes less than 10% of the floor space and most. Teachers received a full day of training, focused on the use of flexible spaces. CG: Traditional desk and chairs. • 8 weeks duration. |
| Bluteau J, et al., 2022 [20] | Canada | NRCT | 6 classrooms (n = 3 intervention) of 5 schools, 107 children (n = 56 intervention) | P | 11.23±0.7 | IG: Flexible seating where some or all the desks were replaced by furniture that offers a variety of work surfaces, seating sizes and heights, body positions. CG: Traditional desk and chairs. • 4 months duration. |
| Byers T, et al., 2014 [29] | Australia | Mixed method (WSCT + qualitative) | 6 classrooms of 1 school, 164 children. • Qualitative study: teachers (n = NR) and students (n = NR) | S | 12–14 Teachers' characteristics: NR | The group of students of each classroom spent one trimester in the traditional classrooms (CG) and the rest of the school year in the 'New generation learning space' (IG) which included the integration of digital and visual technologies and polycentric layout. This focus on flexibility continued with the installation of modular and moveable furniture such as stools, booths, ottomans and portable tables were integrated with existing desks and chairs. This space offered 3 learning modalities: teacher-centered, learner-centered and informal. • 6 months duration. |
| Cardon G, et al., 2004 [18] | Belgium | RCT pilot study | 2 classrooms (n = 1 intervention) of 2 schools, 47 children (n = 22 intervention) | P | 8.8 ±0.6 | IG: The classroom was equipped with ergonomic furniture that allowed varying working postures and contributed to physiologically correct sitting with movement, called dynamic sitting. All tables had an inclinable top with a minimum inclination of 16º. In addition, the classroom had a standing desk, and the classroom was reorganised to make more floor space available for variations in the daily working routine (corner for reclining mats on the floor). CG: Traditional desk and chairs. • 1.5 school years duration. |
| Halidane C, et al., 2023 [21] | Canada | NRCT | 6 classrooms (n = 3 intervention) of 5 schools, 107 children (n = 56 intervention) | P | 11.18±NR | IG: Flexible seating, as opposed to fixed/ traditional seating, has a more student-friendly and open layout, focusing on comfort and seating choice. Students have the opportunity both to choose various seating options (chairs, exercise balls, cushions, and so on) and to choose their workspaces (round tables, single tables, high tables, and so on). CG: Traditional desk and chairs. • Duration: NR. |

*(Continued)*

**Table 1.** (Continued)

| Author/s, year | Country | Study design | Participants | School (Primary/ Secondary) | Age (Mean ± SD), or range | Intervention and comparator |
|---|---|---|---|---|---|---|
| Hinckson EA, 2013 [30] | New Zealand | Mixed method (RCT pilot study and qualitative) | 3 classrooms (n = 2 intervention) of 2 schools, 30 children (n = 23 intervention) • Qualitative study: teachers/ staff (n = 3), students (n = 16) and parents (n = 2) | P | 7–9 Teachers' and parents' characteristics: NR | IG: Traditional desks and chairs were removed from the classrooms. A circular workstation in the middle of the classroom and semicircular stations situated strategically around the central workstation. The standing workstations were adjusted to children's height. Workstations accommodated 4–5 children. Exercise balls, bean bags, and mats were made available for children to sit when tired. CG: Traditional desk and chairs. • 4 weeks duration. |
| Hartikainen J, et al., 2021 [32] | Finland | Cross-sectional | 2 classrooms of 1 school before (n = 83) and after (n = 47) renovation from traditional to OLS from two separate academic years | P | 9–12 | Open Learning Space: a large space (about 70–80 students) with mobile furniture, which afforded multiple options in classroom layout. The student did not have an assigned place, such as a designated desk. Traditional classrooms with desks for each pupil and one teacher responsible for a class of 20–25 pupils. • Duration: NA. |
| Hartikainen J, et al., 2021 [33] | Finland | Cross-sectional | 15 classrooms of 3 schools (one school with OLS), 204 children (n = 66 in OLS) | P | 9–12 | Open Learning Space: a large space (about 70–80 students and 3 teachers) with mobile furniture, which afforded multiple options in classroom layout. The student did not have an assigned place, such as a designated desk. Traditional classrooms with desks for each pupil and one teacher responsible for a class of 20–25 pupils. • Duration: NA. |
| Hartikainen J, et al., 2022 [34] | Finland | Cross-sectional | 15 classrooms of 3 schools (one school with OLS), 191 children (n = 59 in OLS) | P | 9–12 | Open Learning Space: a large space (about 70–80 students and 3 teachers) with mobile furniture, which afforded multiple options in classroom layout. The student did not have an assigned place, such as a designated desk. Traditional classrooms with desks for each pupil and one teacher responsible for a class of 20–25 pupils. • Duration: NA. |
| Hartikainen J, et al., 2023 [35] | Finland | Cross-sectional | 15 classrooms of 3 schools (one school with OLS), 182 children (n = 57 in OLS) | P | 9–12 | Open Learning Space: a large space (about 70–80 students and 3 teachers) with mobile furniture, which afforded multiple options in classroom layout. The student did not have an assigned place, such as a designated desk. Traditional classrooms with desks for each pupil and one teacher responsible for a class of 20–25 pupils. • Duration: NA. |
| Imms W and Byers T, 2017 [31] | Australia | WSCT | 3 classrooms of 1 school, 52 children | S | 12–13 | The group of students of each classroom, accompanied by their teachers, spent one trimester in each spatial condition: mode 1 (basal, CG): traditional classroom with individual tables and chairs arranged towards a teacher's table and a white board at the front of the room; mode 2 (IG1): classroom with 'cluster' table arrangements and other modifications; and mode 3 (IG2): classroom with multiple whiteboards, multiple portable televisions on wheels and a variety of non-traditional furniture. • 1 academic year duration. |

*(Continued)*

**Table 1.** (Continued)

| Author/s, year | Country | Study design | Participants | School (Primary/Secondary) | Age (Mean ± SD), or range | Intervention and comparator |
|---|---|---|---|---|---|---|
| Kariippanon KE, et al., 2018 [39] | Australia | Qualitative | 12 school principals, 35 teachers and 85 students from 8 primary and secondary schools | P, S | 9–17<br>Teachers' characteristics: Different years of experience, from various educational levels, and across all curriculum areas. | 'Flexible Learning Spaces', defined as spaces with a variety of furniture options in a relatively open space that can be configured in different ways to provide a range of learning experiences and opportunities for both individual and collaborative work, using a range of technologies to facilitate personalised teaching and learning.<br>• Duration: NA. |
| Kariippanon KE, et al., 2019 [17] | Australia | School-based cross-over trial | 9 secondary schools (n = NR intervention), n = 191 adolescents (n = NR intervention) | S | 13.2 ± 1.0 | IG: Flexible Learning Spaces were a combination of standard- and double-sized classrooms (83 m2) and incorporated a range of furniture such as grouped tables, standing workstations, ottomans, couches, and write-able tables and walls. The majority of FLS lacked a distinct front of the classroom, with resources including smart-boards and whiteboard walls available around the room. The teaching approach in the FLS was student-centered and group-work focused.<br>CG: Traditional classrooms were a standard single classroom (50 m2) typically contained a desk and chair for each student, arranged in rows or a u-shape facing the front. Students chose their seat upon entering the room and generally remained there during the lesson. The teaching style was primarily teacher-led.<br>• Duration: NR. |
| Kariippanon KE, et al., 2019 [22] | Australia | School-based cross-over trial | 9 secondary schools (n = NR intervention), n = 60 adolescents (n = NR intervention) | S | 13.2 ± 1.0 | IG: Flexible Learning Spaces were a combination of standard- and double-sized classrooms (83 m2) and incorporated a range of furniture such as grouped tables, standing workstations, ottomans, couches, and write-able tables and walls. The majority of FLS lacked a distinct front of the classroom, with resources including smart- boards and whiteboard walls available around the room. Before the study, teachers engaged in a variety of professional development activities, including visits to schools with flexible learning spaces, attending conferences and short courses on designing and teaching in such spaces.<br>CG: Traditional classrooms were a standard single classroom (50 m2) typically contained a desk and chair for each student, arranged in rows or a u-shape facing the front. Students chose their seat upon entering the room and generally remained there during the lesson.<br>• Duration: NR. |

(*Continued*)

**Table 1.** (Continued)

| Author/s, year | Country | Study design | Participants | School (Primary/ Secondary) | Age (Mean ± SD), or range | Intervention and comparator |
|---|---|---|---|---|---|---|
| Lanningham-Foster L, et al., 2008 [16] | USA | WSCT | 1 school, 24 children | P | 10.2 ± 0.6 | The students attended school in three different conditions: CG: traditional school with individual tables and chairs for each pupil, and the seating arrangement was fixed; IG1: 'Standing Classroom' with activity-promoting games, and desks, which encouraged standing; and c) an activity-permissive environment called 'The Neighborhood', which was designed specifically to encourage an active learning environment. It was over 1000 m2 and look liked as village square. The actual "classroom" was a plasticized hockey rink complete with standing desks and vertical, mobile white-boards that allowed for activity-permissive lessons. The Neighborhood also included miniature golf, basketball hoops, indoor soccer, and climbing mazes. Learning tools (notebook computers and video iPods) were provided both in The Neighborhood and in the Standing Classroom.<br>• 12 weeks duration. |
| Olivera-Ortiz Y, 2021 [36] | USA | Qualitative | 35 children from 2 schools with different socio-economic characteristics which moved from an old building to a new building. | P | 9–11 | Classrooms were modernised with natural light and improved teaching features such as writable and magnetic whiteboards. Transparency was created between classrooms and corridors. Library/media centre areas were open, creating a learning common for students rather than a book museum. Flexible collaboration areas were incorporated to allow for small group instruction, instructional interventions, and other learning activities deemed appropriate by the end user.<br>• Duration: NA. |
| Reinius H, et al., 2021 [37] | Finland | Qualitative | 17 children in 1 school, and 2 teachers | P | 8–9<br>Teachers' characteristics: men with 10 years' experience | The workspace design consisted of a large central open area (Agora) furnished with circular sofas for collaborative work, surrounded by three smaller breakout rooms and a teacher's room. Breakout rooms differed in furniture and layout, featuring bean-bag chairs, small desks with exercise balls, or tables and wheeled chairs. Each room was equipped with digital tools, instructional materials, and wireless connections, with glass partitions between them and no traditional teacher desks.<br>• Duration: NA. |

(*Continued*)

**Table 1.** (Continued)

| Author/s, year | Country | Study design | Participants | School (Primary/ Secondary | Age (Mean ± SD), or range | Intervention and comparator |
|---|---|---|---|---|---|---|
| Vidergor HV, 2022 [38] | Israel | Qualitative | 34 teachers in 2 Primary school (n = 20) and one Secondary school (n = 14) | P, S | Teachers' characteristics: both sexes, 21 of them with over 10 years of experience and half graduate + PhD diploma. 41% were also class/subject coordinators and 3 were also principals. | Teachers taught in 3 types of Innovative Learning Spaces: 1. Traditional classrooms with flexible walls and breakout space. 2. Three expansive learning spaces, catering to two grade levels each (grades 1–2, 3–4, and 5–6). Each ILS is furnished with computers, and a single teacher oversees the students' learning activities. Focus on individual learning. 3. The Secondary school repurposed four classrooms into a spacious open area, accommodating multiple classes studying various subjects simultaneously with different teachers. Additionally, the space features an independent study area with a small library and a 'future class' equipped with 3D printers, large screens, and computers for specialized learning. • Duration: NA. |

Abbreviations: CG: control group; FLS: Flexible Learning Spaces; IG: intervention group; ILS: Innovative Learning Spaces; NA: Not applicable due to study design; NR: not reported; NRCT: non-randomised controlled trial; OLS: Open Learnings Space; P = Primary School; RCT: randomized controlled trial; S: Secondary school; USA: United States of America; WSCT: within-subject control trial.

secondary schools [17, 22, 29, 31] and two in a combined primary and secondary school setting [38, 39]. Studies had the following designs: two crossover trials [17, 22], two pilot studies of cluster randomized controlled trials (RCT) [18, 30], four non-RCT [19–21, 28], three within-subject control trials [16, 29, 31], four cross-sectional [32–35] and four qualitative designs [36–39]. Three studies also added a nested qualitative study to the main design [19, 29, 30].

## Characteristics of Active Learning Classroom interventions

Table 1 shows the design details of the interventions. Interventions duration ranged from 1 month to 1.5 school years. Three studies did not report on the duration of the intervention [17, 21, 22]. In all studies, all or most of the traditional classroom tables and chairs were replaced by innovative and versatile furniture such as standing desks, conventional desks and chairs on wheels that allowed for different working configurations (pairs, small groups, large groups), height-adjustable stools, exercise balls, cushions, bean bags, ottomans, etc., such that students could use different working areas and positions in class. In addition, seven interventions included tablets, TVs, smart boards, laptops and vertical whiteboards [16, 17, 22, 29, 31, 37–39]. Most studies indicated that the classroom configuration was 'polycentric' and that the teacher did not occupy a central place in the classroom [16, 28–30, 37, 38]. In one study, teachers received guidance on how to utilize these spaces effectively [28], while two other studies provided recommendations on the most appropriate design and teaching strategies for these environments [17, 22]. Additionally, one study highlighted that the design of these spaces supported three distinct modalities of learning: teacher-centered, learner-centered, and informal approaches [29].

Fig 2 shows a Word Cloud with the key elements of the interventions included in the studies. The larger, highlighted words indicate the elements that have been mentioned most often in the studies.

The main results on the impact of ALCs on the study variables are shown in Table 2.

**Impact of Active Learning Classrooms interventions on**

**Movement behaviours and physical health.** Of the six studies that examined the association or effect of ALCs with movement behaviours [17–19, 30, 34, 35], four reported a decrease in sitting time and an increase in standing time [17–19, 30], three an increase in step time and number of steps [17–19], four an increase in sit-to-stand transitions [17, 32, 34, 35] and two a decrease in the duration of sedentary bouts [17, 34]. In contrast, two studies found an increase in sedentary time among ALCs students [32, 35] and a decrease in the sit-to-stand transitions [30]. Of the five studies [16, 18, 32, 33, 35] that analysed the association or effect of ALCs on PA during school hours, two studies reported an increase in total PA [16, 18] and one in moderate to vigorous PA (MVPA) in 3rd graders [32]. In contrast, schoolchildren attending ALCs accumulated lower levels of light PA [32], total PA [33] and MVPA [35] than students in traditional classrooms. In all studies sedentary behaviour and PA were assessed by accelerometry, except in the study by Cardon et al., where sedentary behaviour was assessed by observation [18].

**Fig 2. Word cloud with key elements of an Active Learning Classroom.**

**Table 2. Aims, outcomes and main results of the included studies.**

| Author/s, year | Aims | Outcome/s | Main results |
|---|---|---|---|
| Aminian S, et al., 2015 [19] | To test the effectiveness of 'dynamic classroom' environment to increase standing and reduce sitting. | • Sitting time, standing time, stepping time and step counts (ActivPAL accelerometer).<br>• Musculoskeletal pain (modified version of the Nordic Musculoskeletal Questionnaire).<br>• Weight, height, WC and BMI.<br>• Practicality, strengths and challenges of the intervention by interview (teachers) and focus group (students). | • ↓Sitting time, ↑standing time, stepping time and step counts.<br>• No back pain was reported in the IG in the final measurement compared with baseline.<br>• No differences in weight, WC and BMI between IG and CG.<br>• Children preferred using height-appropriate standing workstations over sitting desks for their classwork. Teachers appreciated the dynamic classroom environment, noting benefits such as increased space, social interactions, and happier children. Also, they recognized the positive impact of PA on children's health and learning, with some noting improved concentration when using standing workstations |
| Attai S, et al., 2019 [28] | To investigate differences between flexible and traditional furniture and the manner in which flexible furniture impacts students' perceptions of environment and engagement. | • Students' perceptions of the environment in the classrooms (Learning Environment Student Survey).<br>• Comfort of the classroom furniture (questionnaire).<br>• Student movement and aspects of student autonomy by observation. | • ↑Satisfaction, comfort, attentiveness, enjoyment, pleasure in the classroom furniture and opportunities for student autonomy and use of furniture for learning and ↓ distraction in the IG versus CG. |
| Bluteau J, et al., 2022 [20] | To examine the influence of flexible seating on the wellbeing and mental health of elementary school students. | • Wellbeing at school (Liddle and Carter questionnaire).<br>• Mental health (BASC-3 questionnaire). | • ↑Well-being and mental health (<internalising problems, inattention/hyperactivity, and emotional symptoms) in girls in the IG versus CG, but ↓mental health (>internalising problems, inattention/hyperactivity, and emotional symptoms) in boys in the IG versus CG. |
| Byers T, et al., 2014 [29] | To examine the impact of 'new generation learning spaces' on teachers' pedagogy, student engagement and student learning outcomes. | • Student learning experiences and levels of engagement (questionnaire).<br>• Academic achievement (English and mathematics score).<br>• Teacher pedagogy changes and perception of student's engagement by focus group. | • ↑Positive learning experiences and level of engagement in learning of the students in the IG versus CG.<br>• No differences in academic achievement between IG and CG.<br>• From the teachers' perspective, the change in space had a significant and positive impact on both their practice (↑reflection and awareness of the need for change to adapt to space) and the level of student engagement (↑motivation, interest, excitement and mood). |
| Cardon G, et al., 2004 [18] | To evaluate the differences between a traditional school and a 'Moving school' in posture, duration and frequency of sitting in the classroom. | • Duration and frequency of different postures in the classroom (Portable Ergonomic Observation method).<br>• PA levels (CSA accelerometer).<br>• Back and neck pain (questionnaire). | • ↓Time static sitting and this posture is replaced by dynamic sitting (53%), standing (31%) and walking around (10%) in the students of the IG versus CG.<br>• ↑PA in students in the IG versus CG.<br>• ↓Back and neck pain in students in the IG versus CG. |
| Halidane C, et al., 2023 [21] | To compare the quality of interaction between student and teacher, stress and mental health indicators' scores between two groups (flexible seating and fixed seating). | • Mental health (BASC-3 questionnaire).<br>• Stress by nocturnal heart rate variability (Hexoskin® biometric vests).<br>• The quality of student-teacher interaction (Classroom Assessment Scoring system (CLASS)-video recorder method). | • ↑Mental health (<internalising problems, inattention/hyperactivity, school problems, and emotional symptoms) in girls in the IG versus CG. However, ↓in boys in the IG, except for school problems.<br>• No difference in stress between IG and CG.<br>• ↑Quality of student-teacher interaction (emotional support, classroom organization, instructional support) in the IG versus CG. |

(*Continued*)

**Table 2.** (Continued)

| Author/s, year | Aims | Outcome/s | Main results |
|---|---|---|---|
| Hinckson EA, 2013 [30] | To examine the acceptability of introducing standing workstations in elementary-school classrooms; and to quantify changes in children's time sitting, standing, walking, step counts, sit-to-stand transitions and musculoskeletal discomfort. | • Sitting time, standing time, stepping time, step counts and sit-to-stand transitions (ActivPAL accelerometer).<br>• Musculoskeletal pain (Nordic Musculoskeletal Questionnaire).<br>• Children's and parents' reactions to the standing workstations by focus groups. And semi-structured interview with the principal. | • ↓Sitting time and sit-to-stand transitions, and ↑standing time. No differences in stepping time and step counts between IG and CG.<br>• IG reported little or no pain or fatigue.<br>• Children spoke enthusiastically of the standing workstations. Parents noted no noticeable changes in their children's energy level at home. While the 4-grade teacher embraced the new environment, the 3-grade teacher found it distracting. The principal supported the standing workstations, citing their potential for flexibility in learning and health benefits. |
| Hartikainen J, et al., 2021 [32] | To investigate differences between ST, breaks from ST and PA levels of students in grades 3 and 5 in two separate academic years before and after a school renovation (from traditional to OLS). | • ST, BST, and PA (waist triaxial accelerometer). | • ↑ST and ↓light PA, but ↑number of BST in the cohort of 5 graders after renovation.<br>• In 3 grade students, ↑MVPA after renovation. |
| Hartikainen J, et al., 2021 [33] | To investigate the associations between classroom type (OLS vs traditional) with student engagement and PA. | • Behavioural and emotional engagement (questionnaire).<br>• Total PA (waist triaxial accelerometer). | • OLS was associated with better attitude towards school, but not with task-focused behaviour.<br>• Students in conventional classrooms were more physically active than students in OLS. |
| Hartikainen J, et al., 2022 [34] | To compare sedentary bout durations and sit-to-stand transitions between schools with OLS and schools with conventional classrooms. | • Sedentary bouts durations (waist triaxial accelerometer) and sit-to-stand transitions (accelerometer attached on the mid-anterior thigh). | • ↑1-to-4 sedentary bouts/h, ↓+10-min sedentary bouts/h and no differences were observed between groups in 5-to-9 sedentary bouts/h in OLS.<br>• ↑sit-to-stand transitions in OLS. |
| Hartikainen J, et al., 2023 [35] | To investigate the effects of classroom type (OLS vs traditional) on PA; and the associations between teachers' instructions about students' movement and PA in students in grades 3 and 5. | • PA, ST, BST, sedentary bouts and active bouts (waist- triaxial accelerometer).<br>• Teachers' instructions on student movement (systematically observed). | • ↑ST, ↓MVPA in OLS in 5 grades.<br>• ↑the number of BST.<br>• Teachers' instructions regarding 5 graders' movement were more restrictive in OLS, and both 3 and 5 graders had more instructed transitions in OLS. In conventional classrooms, pupils had more teacher-organised PA. |
| Imms W and Byers T, 2017 [31] | To investigate the effect of three different classroom spaces (traditional VS semi open and flexible designs -IG1- and open and flexible designs -IG2-) on perceptions of teaching quality, student engagement and achievement in mathematics. | • Teacher pedagogy (LTPS questionnaire).<br>• Student engagement (LTPS questionnaire).<br>• Performance in mathematics (standardised mathematics scores). | • ↑Student-centred teaching practices, student engagement, and mathematics scores in students in the IG2 (and to a lesser extent in the IG1) compared to traditional space. |
| Kariippanon KE, et al., 2018 [39] | To examine the perceptions of school leadership teams and teachers' and students' 'lived experience' of flexible learning spaces in eight Government schools that had independently made changes to their learning environments. | • Pedagogical approaches employed by teaching by school leadership teams´ open-ended interviews.<br>• The learning experience and the relationship between flexible spaces and the physical, social and emotional wellbeing of teachers and students by focus groups. | -Flexible learning spaces were described as supporting student-centred teaching approaches, encouraging self-regulation, collaboration, autonomy and engagement. Adapted spaces were described as more pleasant, comfortable, inclusive and encouraging greater interaction. |
| Kariippanon KE, et al., 2019 [17] | To measure and compare adolescent sitting patterns between traditional classroom and 'flexible learning spaces'. | • Sitting time, standing time, stepping time, sedentary bouts durations, and the number of BST (ActivPAL accelerometer). | • ↓Sitting time and bouts of prolonged sitting (≤30 min), ↑the number of BST, time standing and stepping time between IG and CG. |
| Kariippanon KE, et al., 2019 [22] | To measure and compare adolescent classroom behaviour between traditional classrooms and 'flexible learning spaces'; and assess the effect of the space and teaching approach. | • Students' in-class behaviour (student level setting, mode of learning, academic behaviour, interactions with peers and teacher, use of technology) by systematically observed. | • ↑time in large group settings, collaborating, interacting with peers, and actively engaged between IG and CG.<br>• ↓time being taught in a whole class setting, engaged in teacher-led instruction, working individually, verbally off-task, and using technology. |

*(Continued)*

**Table 2.** (Continued)

| Author/s, year | Aims | Outcome/s | Main results |
|---|---|---|---|
| Lanningham-Foster L, et al., 2008 [16] | To examine the impact of three different spaces (traditional VS 'Standing Classroom'-IG1- and 'an activity-permissive environment'-IG2-) on PA levels. | • PA levels by accelerometer. | • ↑PA levels in the 'activity-permissive space' more than in the 'Standing Classroom' or traditionally designed classrooms. |
| Olivera-Ortiz Y, 2021 [36] | To explore the attributes of design and spaces that students value and perceive as having an impact on their learning and engagement. | • Students' perceptions of their learning experiences and engagement by focus group and semi-structured interviews. | • A better learning experience is created by a building that is open, inviting, noise efficient and safe. |
| Reinius H, et al., 2021 [37] | To investigate how the new, flexible learning spaces were used by students and teachers; and to examine the association between changes in the physical learning space and collaborative forms of learning and teaching. | • Collaboration activities between pupils and teachers by observation and interviews. | • Students are actively involved in collaborative learning and teaching, often working in pairs or small groups. They embraced mobility in their learning, exercising agency in their choice of study locations and methods.<br>• Teachers agreed that the school environment facilitated collaborative learning and highlighted the importance of professional co-planning between teachers. |
| Vidergor HV, 2022 [38] | To investigate elementary-and middle-school teachers' perceptions of teaching in Innovative Learning Spaces. | • Perception and practice teaching in ILS by semi-structured interview and observation. | • Teachers recognized the necessity of adapting teaching methods and the advantages of personalized learning.<br>• Challenges: organization, lack of professional development, concerns about the suitability of the ILS for students with special needs.<br>• Elementary teachers emphasized challenges related to planning, student assessment, budget constraints, and equipment suitability, while middle school teachers were more focused on concerns about maintaining control and their role as knowledge conveyors. |

↑ = increase; ↓ = decrease; BASC-3: Behaviour Assessment System for Children, Third Edition; BMI: body mass (calculated from weight in kg divided by squared height in cm); BST: breaks from sedentary time; CG: control group; CSA: Computer Science Application; IG: intervention group; ILS: Innovative Learning Space; LTPS: Linking Teaching, Pedagogy and Space questionnaire; MVPA: moderate to vigorous physical activity; OLS: Open Learning Spaces; PA: physical activity; ST: sedentary time; WC: waist circumference.

Anthropometric and adiposity variables were examined in one study where no differences were found between students in traditional and ALCs [19]. Of the four studies that examined pain or discomfort with the intervention furniture [18, 19, 28, 30], two found no differences between students in traditional and alternative classrooms [19, 30] and two found increased comfort [28] or decreased back and neck pain in students attending innovative classrooms compared to students attending traditional classes [18].

**Mental health indicators.** Two studies examined the impact of ALCs interventions on well-being at school [20] and mental health (measured by the BASC-3 questionnaire) [20, 21] in schoolchildren. One showed that girls attending classes with flexible furniture improved their school well-being and mental health (<internalising problems, inattention/hyperactivity, and emotional symptoms) compared to girls who attended traditional classrooms, but these differences were not observed in boys [20]. Similarly, an intervention in which students had the opportunity to choose both seating and workspaces (round tables, single desks, high desks, etc.), was effective in reducing scores for internalising problems, inattention/hyperactivity, school problems, and emotional symptoms in girls. However, no changes were observed in boys, except for an improvement in the school problems dimension [21].

**Academic indicators.** Of the five studies that examined the effect of ALCs on academic indicators, two studies assessed academic performance by school grades or standarised

performance tests [29, 31] and five assessed engagements in learning by questionnaire [28, 29, 31, 33] or by observation [22]. In the study by Byers et al. found no differences in English and Mathematics scores between students in different learning spaces [29]. However, Imms and Byers described an improvement on standardised tests in Mathematics for students in the classroom with whiteboards, non-traditional furniture and multiple portable television compared to adolescents in traditional classrooms [30]. Engagement in learning increased in all studies [22, 28, 29, 31, 33], although in Hartikainen et al. (2021b) study ALC was not associated with task-focused behaviour [33].

## Community's perceptions and teaching practices in Active Learning Classrooms

**Changes in teachers' pedagogical strategies.** Of the seven studies that investigated how innovative learning spaces influence teaching practice [21, 22, 29, 31, 35, 37, 39], three described these spaces as favoring student-centred pedagogy over teacher-centred teaching [22, 31, 39]. In the study by Reinius et al., flexible classrooms were perceived by teachers as opportunities for interactive and collaborative work among teachers, where they engaged in co-teaching, co-planning and sharing experiences [37]. In another study, the authors described an increase in reflection on practice and an awareness of the need to change pedagogical strategies to adapt to new spaces [29], and Halidane et al. found an increase in the quality of interactions between teachers and students [21]. Finally, Hartikainen et al. described that teachers in ALC were more permissive with students' transitions from one space to another, but more restrictive with movement in general, compared to instructions in traditional classrooms. In addition, teachers in traditional classrooms had more organized physical activities in the classroom than in ALC [35].

**Perceptions of teachers, school staff, family and students teaching and learning in Active Learning Classrooms.** Seven studies described the perceptions of teachers, students, parents and staff in ALCs [19, 29, 30, 36–39]. Teachers described the benefits of ALCs as increased concentration [19], motivation, interest, enthusiasm, and mood in students' learning [29, 30, 39], student collaboration and autonomy [37, 38], opportunities to personalize learning [38], and physical benefits [30, 39]. Challenges included distraction for some students [30, 39] or organizational problems, lack of professional development, sustainability, or adapting these spaces for students with special needs [38]. Students described their preference for standing over sitting [19], enthusiasm [30], and more positive learning experiences [36, 39] as advantages of ALCs. Only one study analysed parents' perceptions, which indicated that they did not perceive any changes in pupils´ PA outside of school [30]. Two studies examined the perceptions of school staff regarding ALCs [30, 39], emphasizing their potential to increase learning flexibility and improve students' health [30]. They also highlighted benefits such as enhanced student collaboration, greater engagement, and improved social and emotional well-being for both students and teachers [39].

## Discussion

The primary aim of this scoping review was to comprehensively identify and synthesize the literature on the effect of ALCs on reducing sedentary behaviour, increasing PA levels, promoting well-being, and improving academic indicators in children and adolescents. Our synthesis shows a predominantly positive influence of ALCs on sedentary behaviour, learning engagement and psychological well-being; and mixed results on PA and academic performance; and no effect on adiposity. Furthermore, ALCs have a neutral or positive effect on muscular pain. Our findings also suggest that teachers in ALCs use student-centred and collaborative

pedagogies more than in traditional classrooms. Moreover, innovative learning spaces are positively perceived and well accepted by the entire educational community. However, the paucity of studies and the lack of robust methodological designs, with small sample sizes and heterogeneous interventions and outcomes, make it difficult to draw firm conclusions. Nevertheless, the findings are valuable in guiding researchers and educators in the design of future research studies and the implementation of interventions in school context.

Overall, most of the included studies showed an increase in standing time and a decrease in sitting time, but at the same time no change in step count, step time, or PA. This suggests that interventions to promote flexible furniture and dynamic workspaces may be effective in reducing sedentary behaviour and increasing standing time but may not be sufficient to increase overall PA levels. A possible explanation for the differences in the effect of interventions on increasing PA observed in this review may lie not so much in their design but in the use and dynamics established for working in these spaces. Interventions that implement zone-based routines [18] or specific methodologies promoting student-centered learning while optimizing the space [17] appear to have been more effective than those that did not specify such practices [32, 33]. The physical size of classrooms may also have been a key factor, as larger classrooms might encourage greater PA [16, 17]. Improving the effectiveness of such interventions in increasing PA levels may require the implementation of specific organizational and pedagogical strategies that highlight the optimal use of flexible furniture and spaces. In addition, the inclusion of structured active breaks, whether integrated with curriculum content or not, led by educators, could potentially lead to greater success in promoting PA in educational settings [35]. More studies are needed to confirm this.

Regarding the use of alternative classroom furniture on musculoskeletal pain, two studies found that students in ALCs had less pain [18] and more comfort [28] than the control group, while two other studies found no differences between the groups [19, 30]. Thus, these results, although weak, suggest that there is no clear negative effect of their use, as has been reported in other studies on university students [40]. It should be noted that in this study where users reported discomfort or pain, the intervention consisted of replacing traditional tables and chairs with high desks, where the only option is to stand, sometimes without the possibility of sitting or reclining on a stool. However, ALCs offer different working areas that promote moving around the classroom from one area to another, as well as diverse furniture that allows not only for a transition from sitting to standing but also for alternating sitting postures, which could explain the differences between these studies and the ones included in this scoping review.

Only one study [19] analysed the effect of ALCs on adiposity and found no difference between students in dynamically designed classrooms and students with traditional desks and tables. This non-effect could be explained by the short duration of the intervention (only 9 weeks); however, it is not possible to draw solid conclusions. Further studies should examine the impact of innovative spaces on students' body composition (adiposity and muscle mass) and other cardiometabolic benefits.

Although the evidence on the effect of ALCs on the well-being and mental health of schoolchildren is very limited (only 2 studies), the results are consistent across studies [20, 21]. They show an increase in girls' mental health and a decrease in boys' mental health, with clear sex differences that could be explained by differences in the skills required to learn in open learning spaces between boys and girls. It should be noted that flexible learning spaces require certain skills from students, such as self-control, problem-solving, autonomy, cooperation and working together, etc., which girls may be better at [41], and therefore may not be suitable for all students. This would suggest that learning in such spaces requires a gradual introduction and a period of adjustment, especially for boys or those students with less aptitude for working in such spaces.

In all the studies included in this review, no negative results were found in terms of feasibility or academic indicators such as academic performance, or engagement in learning, except in three studies where teachers described distraction as a barrier for some students [30, 39] or organisational problems, lack of professional development, sustainability, or adapting these spaces for students with special needs [38]. In line with the Kariippanon et al. review [23] our results suggest that ALCs may enhance students' engagement in learning and not be detrimental to academic performance. This positive effect on engagement in learning could be explained by two reasons: 1) the direct relationship between moving more and sitting less with cognitive functioning described in the literature [11]; and/or 2) the fact that ALCs encourage learners to be an active part of their learning [22, 23] and consequently increase their participation in what they are learning. However, to confirm these speculations, RCTs with mediation analyses would be needed to establish causal relationships. On the other hand, in four of the five studies in this review engagement was measured by self-reported [28, 29, 31, 33] or by observation [22]. Therefore, it is crucial that future longitudinal and experimental studies include validated measures of cognition such as executive function and objectively measured academic achievement (e.g. grades in different curriculum areas) as outcome measures to provide direct evidence of their impact on academic indicators.

Our results, stemming from both quantitative and qualitative research, suggest that teachers in ALCs utilise more student-centered pedagogies, invest more time in collaborative activities, and have a better student-teacher relationship than when teaching in traditional classrooms [21, 22, 29, 31, 35, 37, 39], although evidence is very limited given the heterogeneity of variables and instruments used in the studies. More studies are needed to confirm these findings.

Finally, both students and teachers perceive more benefits than barriers in ALCs. However, the studies are very few and heterogeneous in their study variables, so the results should be interpreted with caution. Considering that the feasibility of ALCs depends mostly on teachers and learners, future studies should also include qualitative research to explore teachers' and learners' perceptions and ensure feasibility and scalability.

## Strengths and limitations

Although this study comprehensively reviews, following a rigorous methodology, both the quantitative and qualitative available literature about the influence of ALCs on reducing sedentary behaviour, increasing PA levels, promoting well-being, and improving academic indicators in children and adolescents, the consistency of findings is limited by the small sample of included studies and their methodological drawbacks. Therefore, there is a need for well-designed, long-term clinical trials on the impact of ALCs on physical and mental health and academic indicators in children and adolescents, and on teaching practices. Another limitation of this review lies in the lack of a universally accepted definition of ALCs. As a result, the interventions and activities conducted in these spaces are highly heterogeneous, making it challenging to compare findings across studies. This heterogeneity also limits our ability to propose a standardised approach to the design and use of such classrooms.

## Future directions

Some recommendations from the results of this review include the need to rigorously evaluate different patterns of movement in the classroom, and to determine whether these interventions reduce sedentary time in and out of the classroom. There is also a need to include valid and accepted measures of academic and cognitive performance as outcomes of these interventions, as well as how much of the effect of these ALCs interventions is due to changes in classroom design, and how much is due to changes in the pedagogical strategies associated with

these designs. Finally, none of the studies addressed adherence to interventions, and only 4 of the 19 included studies reported strategies for implementing ALCs. Given the importance of these factors for scalability, future research should focus on addressing these gaps.

## Conclusion

In conclusion, although this review suggests a positive effect of ALCs on children's sedentary behaviour, learning engagement and psychological well-being; and mixed results on PA, physical health and academic performance, the consistency of our results is not robust. Thus, studies using well-designed randomised trials, with larger samples, over a full academic year, and where implementation strategies are well described, are needed.

## Supporting information

**S1 Table. Preferred Reporting Items for Systematic reviews and Meta-Analyses extension for Scoping Reviews (PRISMA-ScR) checklist.**
(DOCX)

**S2 Table. Summary of the search strategy.**
(DOCX)

**S3 Table. Studies excluded after full text read with the reasons for exclusion.**
(DOCX)

## Author Contributions

**Conceptualization:** Mairena Sánchez-López, María Eugenia Visier-Alfonso.

**Investigation:** Mairena Sánchez-López, Jesús Violero-Mellado, Vicente Martínez-Vizcaíno.

**Methodology:** Mairena Sánchez-López, Jesús Violero-Mellado, Vicente Martínez-Vizcaíno, Arto Laukkanen, Arja Sääkslahti, María Eugenia Visier-Alfonso.

**Resources:** Vicente Martínez-Vizcaíno.

**Supervision:** Arto Laukkanen, Arja Sääkslahti.

**Validation:** Mairena Sánchez-López, Jesús Violero-Mellado, María Eugenia Visier-Alfonso.

**Visualization:** Mairena Sánchez-López, Jesús Violero-Mellado, María Eugenia Visier-Alfonso.

**Writing – original draft:** Mairena Sánchez-López.

**Writing – review & editing:** Mairena Sánchez-López, Jesús Violero-Mellado, Vicente Martínez-Vizcaíno, Arto Laukkanen, Arja Sääkslahti, María Eugenia Visier-Alfonso.

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
