## [Decision Letter · Decision Letter 0]

25 Oct 2024

PONE-D-24-40271Impact and perceptions of Active Learning Classroom on reducing sedentary behaviour and improving physical and mental health and academic indicators in children and adolescents: a scoping reviewPLOS ONE

Dear Dr. Sánchez-López,

Thank you for submitting your manuscript to PLOS ONE. After careful consideration, we feel that it has merit but does not fully meet PLOS ONE’s publication criteria as it currently stands. Therefore, we invite you to submit a revised version of the manuscript that addresses the points raised during the review process.

 Please submit your revised manuscript by Dec 09 2024 11:59PM. If you will need more time than this to complete your revisions, please reply to this message or contact the journal office at plosone@plos.org. Please include the following items when submitting your revised manuscript:A rebuttal letter that responds to each point raised by the academic editor and reviewer(s). You should upload this letter as a separate file labeled 'Response to Reviewers'.A marked-up copy of your manuscript that highlights changes made to the original version. You should upload this as a separate file labeled 'Revised Manuscript with Track Changes'.An unmarked version of your revised paper without tracked changes. You should upload this as a separate file labeled 'Manuscript'.If applicable, we recommend that you deposit your laboratory protocols in protocols.io to enhance the reproducibility of your results. Protocols.io assigns your protocol its own identifier (DOI) so that it can be cited independently in the future. For instructions see: https://journals.plos.org/plosone/s/submission-guidelines#loc-laboratory-protocols. Additionally, PLOS ONE offers an option for publishing peer-reviewed Lab Protocol articles, which describe protocols hosted on protocols.io. Read more information on sharing protocols at https://plos.org/protocols?utm_medium=editorial-email&utm_source=authorletters&utm_campaign=protocols.

We look forward to receiving your revised manuscript.

Kind regards,

Heather Macdonald, Ph.D

Academic Editor

PLOS ONE

Journal Requirements:

plos.org/plosone/s/file%3fid=wjVg/PLOSOne_formatting_sample_main_body.pdf%20andWhen submitting your revision, we need you to address these additional requirements.

3. Thank you for stating the following financial disclosure: “This review was conducted while Mairea Sánchez-López was benefiting from a research grant awarded by the Spanish Ministry of Universities (Senior Mobility Grants ‘Salvador Madariaga 2022’, reference number: PRX22/00245).”

4. We note that your Data Availability Statement is currently as follows: “All relevant data are within the manuscript and in Supporting Information files.”

Please confirm at this time whether or not your submission contains all raw data required to replicate the results of your study. Authors must share the “minimal data set” for their submission. PLOS defines the minimal data set to consist of the data required to replicate all study findings reported in the article, as well as related metadata and methods (https://journals.plos.org/plosone/s/data-availability#loc-minimal-data-set-definition). For example, authors should submit the following data: - The values behind the means, standard deviations and other measures reported; - The values used to build graphs; - The points extracted from images for analysis. Authors do not need to submit their entire data set if only a portion of the data was used in the reported study. If your submission does not contain these data, please either upload them as Supporting Information files or deposit them to a stable, public repository and provide us with the relevant URLs, DOIs, or accession numbers. For a list of recommended repositories, please see https://journals.plos.org/plosone/s/recommended-repositories. If there are ethical or legal restrictions on sharing a de-identified data set, please explain them in detail (e.g., data contain potentially sensitive information, data are owned by a third-party organization, etc.) and who has imposed them (e.g., an ethics committee). Please also provide contact information for a data access committee, ethics committee, or other institutional body to which data requests may be sent. If data are owned by a third party, please indicate how others may request data access.

5. As required by our policy on Data Availability, please ensure your manuscript or supplementary information includes the following: A numbered table of all studies identified in the literature search, including those that were excluded from the analyses. For every excluded study, the table should list the reason(s) for exclusion. If any of the included studies are unpublished, include a link (URL) to the primary source or detailed information about how the content can be accessed. A table of all data extracted from the primary research sources for the systematic review and/or meta-analysis. The table must include the following information for each study: Name of data extractors and date of data extraction Confirmation that the study was eligible to be included in the review. All data extracted from each study for the reported systematic review and/or meta-analysis that would be needed to replicate your analyses. If data or supporting information were obtained from another source (e.g. correspondence with the author of the original research article), please provide the source of data and dates on which the data/information were obtained by your research group. If applicable for your analysis, a table showing the completed risk of bias and quality/certainty assessments for each study or outcome. Please ensure this is provided for each domain or parameter assessed. For example, if you used the Cochrane risk-of-bias tool for randomized trials, provide answers to each of the signalling questions for each study. If you used GRADE to assess certainty of evidence, provide judgements about each of the quality of evidence factor. This should be provided for each outcome. An explanation of how missing data were handled. This information can be included in the main text, supplementary information, or relevant data repository. Please note that providing these underlying data is a requirement for publication in this journal, and if these data are not provided your manuscript might be rejected.

Additional Editor Comments:

The manuscript is well-written and provides a thorough review of the literature related to active learning classrooms. In addition to the Reviewers' comments, please consider these additional comments:

1. Title: Should Active Learning Classroom be made plural here as per the rest of the manuscript?

2. Lines 94-97: I find the wording here awkward - please revise. It seems that the review addressed the impact of ALCs on pedagogical strategies, but the objective related to perceptions does not fit with the objective related to the impact of ALCs on... (i.e., it doesn't sense to say "...the impact of ALCs on perceptions of the school community about ALCs...). Perhaps a 2nd objective is needed that focuses on implementation/perceptions of ALCs. In turn, in the results, perceptions would not fall under the subheading related to Impact of ALC interventions

4. Lines 115-124: The wording of this list could be improved, particularly on lines 120-121 - reported at least one "physical" and "mental" doesn't make sense. Also, the last part of the sentence regarding methods could be a separate sentence. 

5.. Line 125: Change to "Studies were included if they: focused on higher education..." and then ensure that the text matches for each exclusion criteria (i.e., change "include" to "included" on Line 130. 

6. Line 136: Consider starting a new sentence with "Studies were screened..."

7. Lines 154-156: Consider changing the language here to say that 1084 publications were screened, and then 18 eligible publications were included. Please also clarify if the 18 publications represented 18 independent studies.

8. Line 253: Change Active Learning Space to Active Learning Classroom.

9. Line 318: Feasibility is mentioned here but not in the methods and I don't see it listed in the search terms in the supplementary table. It would be interesting to know how many studies reported feasibility indicators such as adherence. Similarly, in the last paragraph of the paper, the authors mention implementation strategies, yet strategies are only mentioned once in the discussion and implementation/implementation strategies were not included in the search strategy. This would be valuable information to consider.  

Reviewers' comments:

Reviewer's Responses to Questions

**Comments to the Author**

1. Is the manuscript technically sound, and do the data support the conclusions?

Reviewer #1: Yes

Reviewer #2: Yes

2. Has the statistical analysis been performed appropriately and rigorously? 

Reviewer #1: N/A

Reviewer #2: Yes

3. Have the authors made all data underlying the findings in their manuscript fully available?

Reviewer #1: Yes

Reviewer #2: Yes

4. Is the manuscript presented in an intelligible fashion and written in standard English?

Reviewer #1: No

Reviewer #2: Yes

5. Review Comments to the Author

Reviewer #1: Dear Authors,

Overall, the manuscript is informative and reviews research on Active Learning Classroom (ALC) and various behavioural, health and academic indicators in children and adolescents in the school setting. The comments and questions below highlight points that need further attention and work.

Comments:

1) Introduction, page 1. Although ALC lacks a universally accepted definition, I recommend explaining ‘active learning’ somewhat more clearly (line 68). Does the term only refer to learning that occurs while students are moving? Or is it something more than that?

2) Material and Methods, page 5. You write that the search was limited to studies published in English. Later in the paragraph, you write that you also search for studies in Spanish. What is the conclusion from this? Did you not find any studies published in Spanish? Please explain or correct. Now it seems a bit contradictory.

3) Material and Methods, page 7. The Data analysis and synthesis lack clarity. If included, please briefly explain how the data analysis was conducted. In addition, it is better to only refer to the PRISMA flow chart at the beginning of the Results section.

4) Discussion, page 27. You suggest using objectively measured academic achievement as outcome measures. Please provide suggestions for some measures. Are you thinking of subject scores and grades or other measures? May be of importance to guide future research on ALC.

5) Strengths and Limitations, page 28. It is positive that you describe the non-unanimous definitions of ALC as a limitation. However, I think that the actual heterogeneity in ALC interventions and activities also is a challenge when comparing different studies. Please add this issue when describing the possibility of recommending the design and use of classrooms.

6) Table 1. Please review the right column describing ‘Intervention and comparator’. For example, the description of the Attai et al. study is not complete. Report duration in a consistent way. For example, add to the Byers et al. study. Also, the Imms and Byers study is misplaced in alphabetical order. Please list the abbreviations at the end in alphabetic or any other logical order. Now the abbreviations are in adhoc order. Please correct typos ‘range’.

7) Table 2. The Imms and Byers study is misplaced in alphabetical order. Please list the abbreviations at the end in alphabetic or any other logical order. For example, in alphabetic order, sorted by the Outcome and Main results categories. Correct typos. For example, the abbreviation IG is written as GI several times.

I have no concerns about dual publication, research ethics, or publication ethics.

Reviewer #2: This scoping review addresses an important and timely topic in the field of physical activity and education for children and adolescents (3-18 years). The authors aim to synthesize the available literature on the impact of Active Learning Classrooms (ALCs) on reducing sedentary behavior, increasing physical activity, and improving physical health, mental health, and academic indicators in school-aged children. Additionally, they explore the perceptions of the educational community and teaching practices associated with these learning spaces.

The review followed a rigorous methodology, adhering to the Joanna Briggs Methods and PRISMA guidelines for scoping reviews. The authors conducted a comprehensive search across multiple databases, including MEDLINE, ERIC, SCOPUS, and ProQuest Education, focusing on peer-reviewed studies published in English up to 2023.

This scoping review makes a significant contribution to the field by synthesizing the current state of knowledge on ALCs and their impacts on children's health and learning. Despite the heterogeneity of interventions and some methodological limitations, the findings suggest that ALCs may offer a promising approach to reducing sedentary behavior and improving various health and educational outcomes for children and adolescents.

Minor Revisions Suggested:

1. Definition of Active Learning Classrooms (ALCs)

The review would benefit from a clearer definition of ALCs in the introduction or methodology section. While the characteristics are well-described in the results, a more explicit explanation of what constitutes an ALC during the screening process would enhance clarity. The review acknowledges the heterogeneity of ALC interventions, which is appropriate for a scoping review. However, a more detailed description of the range of interventions included would provide valuable context.

2. Word Cloud Methodology: The pertinence of the word cloud and its methodology should be better explained, particularly in how it addresses the perceptions of the school community about ALCs and teaching practices in these spaces.

3. Technical Jargon:

Define 'MVPA' (Moderate to Vigorous Physical Activity) and 'LPA' (Light Physical Activity) when first introduced in the text, ideally. "Accelerometry" should also be briefly described for readers unfamiliar with the term. in the introduction alongside the concept of sedentary behavior.

4. Resluts Presentation:

In the section on Movement Behaviors and Physical Health, it is stated, 'In contrast, a decrease was reported in LPA, total PA, and MVPA in students attending ALCs [31,32,34],' but the results table does not present data on LPA.

5. Comparative Discussion: The discussion section could be enhanced by comparing the results with other types of interventions aimed at increasing physical activity levels. This could include exploring whether other interventions lead to different outcomes and analyzing the potential causes for the observed effects in ALCs.

6. PLOS authors have the option to publish the peer review history of their article (what does this mean?). If published, this will include your full peer review and any attached files.

Reviewer #1: No

Reviewer #2: **Yes: **Laurie Simard

---

## [Author Response · Author response to Decision Letter 0]

4 Dec 2024

Heather Macdonald, Ph.D

Academic Editor

PLOS ONE

Dear Editor,

Please find enclosed a revised version of our manuscript, “Impact and perceptions of Active Learning Classrooms on reducing sedentary behaviour and improving physical and mental health and academic indicators in children and adolescents: a scoping review” (PONE-D-24-40271). We would like to thank you for giving us the opportunity to revise and improve our manuscript; we also thank the reviewers for the thoughtful and constructive comments. We have considered all the suggestions and have incorporated them into the revised manuscript ('Revised Manuscript with Track Changes'), and as a result, we believe our manuscript is stronger. An itemized point-by-point response to the academic editor and reviewers’ comments is shown below. 

Thank you for your attention to this matter. 

Additional Editor Comments:

The manuscript is well-written and provides a thorough review of the literature related to active learning classrooms. In addition to the Reviewers' comments, please consider these additional comments:

1.Title: Should Active Learning Classroom be made plural here as per the rest of the manuscript?

Author’s response: We agree. We have changed the title to make it plural. Thank you.

2. Lines 94-97: I find the wording here awkward - please revise. It seems that the review addressed the impact of ALCs on pedagogical strategies, but the objective related to perceptions does not fit with the objective related to the impact of ALCs on... (i.e., it doesn't sense to say "...the impact of ALCs on perceptions of the school community about ALCs...). Perhaps a 2nd objective is needed that focuses on implementation/perceptions of ALCs. In turn, in the results, perceptions would not fall under the subheading related to Impact of ALC interventions.

Author’s response: Thank you for the comment. We have reformulated the objectives as follows:

The objectives of this scoping review were to: (i) synthesise the existing literature on the impact of ALCs on reducing sedentary behaviour, increasing PA, well-being and academic indicators in children and adolescents; and (ii) describe the educational community's perceptions and teaching practices used in these learning spaces.

In addition, in the Results section we have added the following subheading to separate the results corresponding to objective (ii): 

Community's perceptions and teaching practices in Active Learning Classrooms

4. Lines 115-124: The wording of this list could be improved, particularly on lines 120-121 - reported at least one "physical" and "mental" doesn't make sense. Also, the last part of the sentence regarding methods could be a separate sentence. 

Author’s response: Thank you. We have reworded the paragraph to make the wording clearer:

-In Material and Methods- Eligibility and Exclusion Criteria:

Studies were included if they met the following criteria:

(i) were conducted in pre-school, primary, or secondary school classrooms (age range: 3–18 years) within standard classroom settings;

(ii) reported on the impact of ALCs in reducing sedentary behaviour and improving the physical and mental health and academic indicators of students, with at least one variable reported in the following categories: movement behaviour (e.g., physical activity, sedentary time, standing, and sitting time); physical health (e.g., adiposity, pain); mental health (e.g., psychological well-being, stress); or academic indicators (e.g., grades, cognition, engagement). In this review an ALC refers to educational environments designed to facilitate not only physical movement within the classroom, but also active student-centred participation, even if unplanned. These spaces often include flexible and non-traditional furniture arrangements that support a variety of collaborative and individualised work configurations.

(iii) examined the perceptions of students, teachers, families, or staff regarding ALCs, including aspects such as acceptability, perceived barriers and facilitators, or the teaching practices employed in these learning spaces.

Quantitative, qualitative and mixed methods study design were included.

5. Line 125: Change to "Studies were included if they: focused on higher education..." and then ensure that the text matches for each exclusion criteria (i.e., change "include" to "included" on Line 130. 

Author’s response: Thank you for your appreciation. We have modified the paragraph as follows:

-In Material and Methods- Eligibility and Exclusion Criteria.

Studies were excluded if they met any of the following criteria:

(i) focused on higher education (e.g., universities) or mixed groups (e.g., school-aged children or adolescents and adults);

(ii) involved special population groups (e.g., children with clinically significant behavioral disorders, such as attention difficulties, or children with overweight/obesity);

(iii) investigated aspects of the built environment in the classroom, including factors such as temperature, light, color, noise, or overall environmental quality;

(iv) included interventions that did not align with the definition of ALC established in this review, as well as those solely involving the replacement of traditional desks with standing desks, fit balls, or pedal desks. These interventions were excluded because they primarily represent changes in posture during classes and do not necessarily modify classroom configuration or teaching-learning dynamics, even if such changes occur incidentally;

(v) full-text access was unavailable, and the authors did not respond to requests for additional data or the complete text;

(vi) were categorized as text or opinion articles, conference abstracts, doctoral theses, dissertations, or review articles.

6. Line 136: Consider starting a new sentence with "Studies were screened..."

Author’s response: Done. Thank you.

7. Lines 154-156: Consider changing the language here to say that 1084 publications were screened, and then 18 eligible publications were included. Please also clarify if the 18 publications represented 18 independent studies.

Author’s response: Thank you for your suggestion. We have rephrased the paragraph as follows:

The electronic search identified 1352 references. After removal duplicates, 1004 studies were screened, and then 19 eligible publications were included [15–21,27–38], which represented 19 independent studies. The search results and the selection process of studies are presented in a flow chart (Fig 1). Studies excluded after reading the full text with reasons for exclusion are shown in supplementary information (S2 Table).

8. Line 253: Change Active Learning Space to Active Learning Classroom.

Author’s response: Done. Thank you.

9. Line 318: Feasibility is mentioned here but not in the methods and I don't see it listed in the search terms in the supplementary table. It would be interesting to know how many studies reported feasibility indicators such as adherence. Similarly, in the last paragraph of the paper, the authors mention implementation strategies, yet strategies are only mentioned once in the discussion and implementation/implementation strategies were not included in the search strategy. This would be valuable information to consider. 

Author’s response: Thank you for your valuable comments. We have repeated the search by including these two terms (feasibility and ‘implementation strategies’) in the search strategy. We have located one more article that meets the inclusion criteria. The reference has also been added to the References section.

Kariippanon KE, Cliff DP, Lancaster SL, Okely AD, Parrish AM. Perceived interplay between flexible learning spaces and teaching, learning and student wellbeing. Learning Environ Res. 2018; 21, 301–320. Doi: 10.1007/s10984-017-9254-9

Consequently, we have modified table S1 (search strategy) and the flow chart (Fig 1). And several parts of the manuscript. Additionally, we have modified the following paragraph in the Results section and added the following one in Future Directions (in Discussion) to better address the topic:

-Characteristics of Active Learning Classroom interventions (in Results)

In one study, teachers received guidance on how to utilize these spaces effectively [27], while two other studies provided recommendations on the most appropriate design and teaching strategies for these environments [16, 21]. Additionally, one study highlighted that the design of these spaces supported three distinct modalities of learning: teacher-centered, learner-centered, and informal approaches.

-In Future Directions (in Discussion)

Finally, none of the studies addressed adherence to interventions, and only four of the 19 included studies reported strategies for implementing ACL. Given the importance of these factors for scalability, future research should focus on addressing these gaps.

5. Review Comments to the Author

Reviewer #1: 

Dear Authors,

Overall, the manuscript is informative and reviews research on Active Learning Classroom (ALC) and various behavioural, health and academic indicators in children and adolescents in the school setting. The comments and questions below highlight points that need further attention and work.

Comments:

1) Introduction, page 1. Although ALC lacks a universally accepted definition, I recommend explaining ‘active learning’ somewhat more clearly (line 68). Does the term only refer to learning that occurs while students are moving? Or is it something more than that?

Author’s response: We thank the reviewer for this valuable comment, which has helped us to clarify this point. As noted in the introduction, there is no universal definition for this term; however, most studies converge on the understanding that open, flexible or active learning environments differ from traditional classroom settings (e.g. rows of desks facing forward). Instead, these environments are characterised by flexible furniture and spatial arrangements that support movement within classroom while encouraging student-centred learning. Following the reviewer's suggestion, we have redrafted the paragraph to improve the definition of ACLs:

-In Introduction section:

Although it is challenging to find a universally accepted definition for this term, in this review, ALC refers to educational environments designed to facilitate not only physical movement within the classroom, but also active student-centred participation, even if unplanned. These spaces typically include flexible, non-traditional furniture arrangements that support various collaborative and individualized working configurations, encouraging students to take an active role in their learning [14].

-In Material and Methods- Eligibility and Exclusion Criteria:

Studies were included if they met the following criteria:

(i) were conducted in pre-school, primary, or secondary school classrooms (age range: 3–18 years) within standard classroom settings;

(ii) reported on the impact of ALCs in reducing sedentary behaviour and improving the physical and mental health and academic indicators of students, with at least one variable reported in the following categories: movement behaviour (e.g., physical activity, sedentary time, standing, and sitting time); physical health (e.g., adiposity, pain); mental health (e.g., psychological well-being, stress); or academic indicators (e.g., grades, cognition, engagement). In this review an ALC refers to educational environments designed to facilitate not only physical movement within the classroom, but also active student-centred participation, even if unplanned. These spaces often include flexible and non-traditional furniture arrangements that support a variety of collaborative and individualised work configurations.

(iii) examined the perceptions of students, teachers, families, or staff regarding ALCs, including aspects such as acceptability, perceived barriers and facilitators, or the teaching practices employed in these learning spaces.

Quantitative, qualitative and mixed methods study design were included.

-In Material and Methods- Eligibility and Exclusion Criteria.

Studies were excluded if they met any of the following criteria:

(i) focused on higher education (e.g., universities) or mixed groups (e.g., school-aged children or adolescents and adults);

(ii) involved special population groups (e.g., children with clinically significant behavioral disorders, such as attention difficulties, or children with overweight/obesity);

(iii) investigated aspects of the built environment in the classroom, including factors such as temperature, light, color, noise, or overall environmental quality;

(iv) included interventions that did not align with the definition of ALC established in this review, as well as those solely involving the replacement of traditional desks with standing desks, fit balls, or pedal desks. These interventions were excluded because they primarily represent changes in posture during classes and do not necessarily modify classroom configuration or teaching-learning dynamics, even if such changes occur incidentally;

(v) full-text access was unavailable, and the authors did not respond to requests for additional data or the complete text;

(vi) were categorized as text or opinion articles, conference abstracts, doctoral theses, dissertations, or review articles.

2) Material and Methods, page 5. You write that the search was limited to studies published in English. Later in the paragraph, you write that you also search for studies in Spanish. What is the conclusion from this? Did you not find any studies published in Spanish? Please explain or correct. Now it seems a bit contradictory.

Author’s response: The reviewer is right. The search was restricted to peer-reviewed articles published in English. We have corrected the error. We apologize for the inconvenience.

3) Material and Methods, page 7. The Data analysis and synthesis lack clarity. If included, please briefly explain how the data analysis was conducted. In addition, it is better to only refer to the PRISMA flow chart at the beginning of the Results section.

Author’s response: We appreciate the reviewer's comment. To clarify how data analysis and synthesis were conducted in this review, we have redrafted the paragraph:

A narrative synthesis was conducted to summarize the results, and the main findings are presented in evidence tables. Additionally, we performed a Word Cloud figure to provide a visual representation of text in which words are displayed in different sizes and colours according to their frequency in the dataset. The words that appear in larger sizes and brighter colours represent the terms that are repeated in the studies to define the ALCs.

4) Discussion, page 27. You suggest using objectively measured academic achievement as outcome measures. Please provide suggestions for some measures. Are you thinking of subject scores and grades or other measures? May be of importance to guide future research on ALC.

Author’s response: Thanks to the reviewer, we have specified what the reviewer commented.

Therefore, it is crucial that the future longitudinal and experimental studies would include validated measures of cognition such as executive function cognition and objectively measured academic achievement (e.g. grades in different curriculum areas) as outcome measures to provide direct evidence of their impact on academic indicators

5) Strengths and Limitations, page 28. It is positive that you describe the non-unanimous definitions of ALC as a limitation. However, I think that the actual heterogeneity in ALC interventions and activities also is a challenge when comparing different studies. Please add this issue when describing the possibility of recommending the design and use of classrooms.

Author’s response: We agree with the reviewer, thus we have added this relevant question.

Another imitation of this review lies in the lack of a universally accepted definition of ALC. As a result, the interventions and activities conducted in these spaces are highly heterogeneous, making it challenging to compare findings across studies. This heterogeneity also limits our ability to propose a standardized approach to the design and use of such classrooms.

6) Table 1. Please review the r

---

## [Decision Letter · Decision Letter 1]

16 Dec 2024

PONE-D-24-40271R1Impact and perceptions of Active Learning Classrooms on reducing sedentary behaviour and improving physical and mental health and academic indicators in children and adolescents: a scoping reviewPLOS ONE

Dear Dr. Sánchez-López,

Thank you for submitting your revised manuscript to PLOS ONE. While most comments were addressed in this revision, Reviewer 1 has additional comments that require your attention. 

We look forward to receiving your revised manuscript.

Kind regards,

Heather Macdonald, Ph.D

Academic Editor

PLOS ONE

Journal Requirements:

Reviewers' comments:

Reviewer's Responses to Questions

**Comments to the Author**

1. If the authors have adequately addressed your comments raised in a previous round of review and you feel that this manuscript is now acceptable for publication, you may indicate that here to bypass the “Comments to the Author” section, enter your conflict of interest statement in the “Confidential to Editor” section, and submit your "Accept" recommendation.

Reviewer #1: (No Response)

Reviewer #2: All comments have been addressed

2. Is the manuscript technically sound, and do the data support the conclusions?

Reviewer #1: Yes

Reviewer #2: Yes

3. Has the statistical analysis been performed appropriately and rigorously? 

Reviewer #1: N/A

Reviewer #2: Yes

4. Have the authors made all data underlying the findings in their manuscript fully available?

Reviewer #1: Yes

Reviewer #2: Yes

5. Is the manuscript presented in an intelligible fashion and written in standard English?

Reviewer #1: No

Reviewer #2: Yes

6. Review Comments to the Author

Reviewer #1: Dear Authors,

After reading the revised manuscript, I believe the revisions have improved it. I am satisfied with your revisions to my previous comments. I also think that the revised objectives helped clarify the intentions of the review. I have a few detailed comments, and I wish you good luck finalising the manuscript.

Detailed comments:

Lines 32-35. I find the wording a bit awkward. It is easier to write ‘…learning spaces (ALCs) impacts movement patterns, physical or mental health…’ or something similar.

Lines 37-38. Please change ‘Eighteen studies’ to Nineteen studies and adjust the count regarding study designs.

Line 107. For clarity, you might consider changing ‘these learning spaces’ to ALCs. A change here also involves a change in the Abstract.

Lines 215-216. I recommend placing Australia before Finland and Belgium before Israel to follow alphabetic order when describing the number of included studies.

Line 240. I think the reference [28] should be included at the end of the sentence.

Line 365. I recommend that you write ‘The size of classrooms…’ as class size is potentially misleading and might be interpreted as the number of students in each class.

Finally, I recommend that the authors carefully proofread the manuscript.

Reviewer #2: I have reviewed the revised manuscript titled "Impact and perceptions of Active Learning Classrooms on reducing sedentary behaviour and improving physical and mental health and academic indicators in

children and adolescents: a scoping review.

I am pleased to report that the authors have adequately addressed the concerns and suggestions raised in my previous review. The revisions have strengthened the manuscript, and I believe it now meets the publication standards of PLOS ONE.

7. PLOS authors have the option to publish the peer review history of their article (what does this mean?). If published, this will include your full peer review and any attached files.

Reviewer #1: No

Reviewer #2: **Yes: **Laurie Simard

---

## [Author Response · Author response to Decision Letter 1]

17 Dec 2024

Heather Macdonald, Ph.D

Academic Editor

PLOS ONE

Dear Editor,

Please find enclosed a revised version of our manuscript, “Impact and perceptions of Active Learning Classrooms on reducing sedentary behaviour and improving physical and mental health and academic indicators in children and adolescents: a scoping review” (PONE-D-24-40271R1). Once again, we would like to thank you for the opportunity to review and improve our manuscript and the reviewers for their thoughtful and constructive comments. We have taken all suggestions into account and incorporated them into the revised manuscript ('Revised Manuscript with Track Changes') and believe that our manuscript is stronger as a result. Below is a point-by-point response to reviewer 1's comments. 

Thank you for your attention to this matter. 

Review Comments to the Author

Reviewer #1: Dear Authors,

After reading the revised manuscript, I believe the revisions have improved it. I am satisfied with your revisions to my previous comments. I also think that the revised objectives helped clarify the intentions of the review. I have a few detailed comments, and I wish you good luck finalising the manuscript.

Detailed comments:

Lines 32-35. I find the wording a bit awkward. It is easier to write ‘…learning spaces (ALCs) impacts movement patterns, physical or mental health…’ or something similar.

Author’s response: Thank you to the reviewer for your valuable comment. We have revised the sentence to make it simpler and more straightforward, as suggested.

“…We searched for peer-reviewed quantitative and qualitative studies published in English that examined the impact of ALCs on movement patterns, physical or mental health, and academic indicators in children and adolescents, as well as those that explored the perceptions of members of the educational community and the teaching practices used in ALCs…”

Lines 37-38. Please change ‘Eighteen studies’ to Nineteen studies and adjust the count regarding study designs.

Author’s response: Done. Thank you.

Line 107. For clarity, you might consider changing ‘these learning spaces’ to ALCs. A change here also involves a change in the Abstract.

Author’s response: Changed in both sections.

Lines 215-216. I recommend placing Australia before Finland and Belgium before Israel to follow alphabetic order when describing the number of included studies.

Author’s response: Done. Thanks for the recommendation.

Line 240. I think the reference [28] should be included at the end of the sentence.

Author’s response: It is true. We have added it. Thank you very much.

Line 365. I recommend that you write ‘The size of classrooms…’ as class size is potentially misleading and might be interpreted as the number of students in each class.

Author’s response: We completely agree. We have changed it. Thank you for your comments.

Finally, I recommend that the authors carefully proofread the manuscript.

Author’s response: The reviewer’s valuable comment is appreciated, as well as their dedication to the review process. The manuscript has been carefully proofread to ensure clarity and accuracy.

Reviewer #2: I have reviewed the revised manuscript titled "Impact and perceptions of Active Learning Classrooms on reducing sedentary behaviour and improving physical and mental health and academic indicators in children and adolescents: a scoping review.

I am pleased to report that the authors have adequately addressed the concerns and suggestions raised in my previous review. The revisions have strengthened the manuscript, and I believe it now meets the publication standards of PLOS ONE.

Author’s response: We appreciate the reviewer's comment.

---

## [Editor Report · Decision Letter 2]

23 Dec 2024

PONE-D-24-40271R2Impact and perceptions of Active Learning Classrooms on reducing sedentary behaviour and improving physical and mental health and academic indicators in children and adolescents: a scoping reviewPLOS ONE

Dear Dr. Sánchez-López,

Thank you for submitting your revised manuscript and addressing the reviewer's comments. Although you noted that the manuscript was proofread, I noted quite a few typographical and grammatical errors throughout, which I listed below. Please note that PLOS ONE does not copyedit manuscripts before publication. 

We look forward to receiving your revised manuscript.

Kind regards,

Heather Macdonald, Ph.D

Academic Editor

PLOS ONE

Journal Requirements:

Additional Editor Comments:

Thank you for addressing the 2nd round of comments. Although the authors noted that they proofread the manuscript in response to the reviewer's request, I note a few final typographical/grammatical errors that should be corrected. Please note that PLOS One does not copyedit accepted manuscripts before publication so the authors may wish to consider professional language and/or copy editing services.

1. Spelling of behavior/behaviour is inconsistent throughout the manuscript. Please use Find/Replace to use one spelling throughout.

2. Line 38: Add "children's" before sedentary

3. Line 61: Remove "the" before the percentage range.

4. Line 90: Does "none of them" refer to the 2 reviews mentioned above? If so, change to "neither review", and use a comma to join the following sentence (...mental health, nor did these reviews answer...)

5. Line 196: Change "of them" to "studies" - same on line 202 - change "of them" to "studies"

6. The text in the 4th column of Table 2 appears to be right justified and this format isn't used in the other tables - please modify

7. Line 288: Change describe to described

8. Line 291: Change include to included

9. Line 308: Please modify the sentence so it doesn't start with "And"

10. Line 314: Add "included" before studies. Also, "but sometimes" seems rather vague and informal

11. Line 323: Consider adding "physical" before size.

12. Line 331: Consider changing to "...students in ALCs"

13. Line 334: Please correct the grammar in this sentence as "...reported in other study in university students" is poorly worded

14. Line 338: "...different furniture that allows not only to vary..." does not make sense - please modify

15. Line 350: Did this study actually assess the social construct of gender? If not, please change this to sex differences

16. Line 353: Remove "in" before "which"

17. Line 359: Change to "teachers described distraction as a barrier for some students"

18. Line 361: Add "the" before Kariipanon

19. Line 362: Should "active learning spaces" be changed to ALCs?

20. Line 365: Reference 9 cited here has round brackets whereas other references are in square brackets

21. Line 369: Change "as self-reported" to "by self-report"

22. Line 370: Remove "would"

23. Line 382: Remove "use" after ALCs

24. Line 389: Consider changing "their findings" to "study findings" or just say "findings"

25. Line 402: Change "ALCs based interventions" to ALC interventions

26. Line 405: Change "four" to 4

27. Line 409: Consider adding "children's" before sedentary

---

## [Author Response · Author response to Decision Letter 2]

3 Jan 2025

Heather Macdonald, Ph.D

Academic Editor

PLOS ONE

Dear Editor,

Please find enclosed a revised version of our manuscript, “Impact and perceptions of Active Learning Classrooms on reducing sedentary behaviour and improving physical and mental health and academic indicators in children and adolescents: a scoping review” (PONE-D-24-40271R2). Once again, we would like to thank you for the opportunity to review and improve our manuscript. We have taken all suggestions into account and incorporated them into the revised manuscript ('Revised Manuscript with Track Changes'). Below is a point-by-point response to comments. 

Thank you for your attention to this matter. 

Response to additional Editor Comments:

Thank you for addressing the 2nd round of comments. Although the authors noted that they proofread the manuscript in response to the reviewer's request, I note a few final typographical/grammatical errors that should be corrected. Please note that PLOS One does not copyedit accepted manuscripts before publication so the authors may wish to consider professional language and/or copy editing services.

Author’s response: The editor’s valuable comment is appreciated, as well as their dedication to the review process. 

1. Spelling of behavior/behaviour is inconsistent throughout the manuscript. Please use Find/Replace to use one spelling throughout.

Author’s response: Done, except in the references section where we have kept the word in American English to respect the titles of the journals where they are published. Thank you.

2. Line 38: Add "children's" before sedentary

Author’s response: Done

3. Line 61: Remove "the" before the percentage range.

Author’s response: Done

4. Line 90: Does "none of them" refer to the 2 reviews mentioned above? If so, change to "neither review", and use a comma to join the following sentence (...mental health, nor did these reviews answer...)

Author’s response: Yes, we have made the suggested changes.

5. Line 196: Change "of them" to "studies" - same on line 202 - change "of them" to "studies"

Author’s response: Done

6. The text in the 4th column of Table 2 appears to be right justified and this format isn't used in the other tables - please modify

Author’s response: Done

7. Line 288: Change describe to described

Author’s response: Done

8. Line 291: Change include to included

Author’s response: Done

9. Line 308: Please modify the sentence so it doesn't start with "And"

Author’s response: Done

10. Line 314: Add "included" before studies. Also, "but sometimes" seems rather vague and informal

Author’s response: Done. We have rephrased the phrase as follows:

“Overall, most of the included studies showed an increase in standing time and a decrease in sitting time, but at the same time no change in step count, step time, or PA”

11. Line 323: Consider adding "physical" before size.

Author’s response: Done. Thanks for the recommendation.

12. Line 331: Consider changing to "...students in ALCs"

Author’s response: We agree. Done. Thank you for your comments.

13. Line 334: Please correct the grammar in this sentence as "...reported in other study in university students" is poorly worded

Author’s response: Done

14. Line 338: "...different furniture that allows not only to vary..." does not make sense - please modify

Author’s response: Done. We have rephrased the phrase as follows: “…, as well as diverse furniture that allows not only for a transition from sitting to standing but also for alternating sitting postures, which could explain the...”

15. Line 350: Did this study actually assess the social construct of gender? If not, please change this to sex differences

Author’s response: Thank you for this appreciation; it refers to sex differences. We have changed it. Thanks.

16. Line 353: Remove "in" before "which"

Author’s response: Done

17. Line 359: Change to "teachers described distraction as a barrier for some students"

Author’s response: Done

18. Line 361: Add "the" before Kariipanon

Author’s response: Done

19. Line 362: Should "active learning spaces" be changed to ALCs?

Author’s response: Yes, that is right. Changed. Thank you

20. Line 365: Reference 9 cited here has round brackets whereas other references are in square brackets

Author’s response: Changed. Thank you

21. Line 369: Change "as self-reported" to "by self-report"

Author’s response: Done

22. Line 370: Remove "would"

Author’s response: Done

23. Line 382: Remove "use" after ALCs

Author’s response: Done

24. Line 389: Consider changing "their findings" to "study findings" or just say "findings"

Author’s response: Thank you for your comment. We have decided to only leave “findings”

25. Line 402: Change "ALCs based interventions" to ALC interventions

Author’s response: Done

26. Line 405: Change "four" to 4

Author’s response: Done

27. Line 409: Consider adding "children's" before sedentary

Author’s response: Done. We have added “Children´s”.

---

## [Editor Report · Decision Letter 3]

8 Jan 2025

Impact and perceptions of Active Learning Classrooms on reducing sedentary behaviour and improving physical and mental health and academic indicators in children and adolescents: a scoping review

PONE-D-24-40271R3

Dear Dr. Sánchez-López,

We’re pleased to inform you that your manuscript has been judged scientifically suitable for publication and will be formally accepted for publication once it meets all outstanding technical requirements.

Kind regards,

Heather Macdonald, Ph.D

Academic Editor

PLOS ONE

Additional Editor Comments (optional):

Thank you for making the requested changes to the manuscript.
---

## [Editor Report · Acceptance letter]

15 Jan 2025

PONE-D-24-40271R3 

PLOS ONE

Dear Dr. Sánchez-López, 

I'm pleased to inform you that your manuscript has been deemed suitable for publication in PLOS ONE. Congratulations! Your manuscript is now being handed over to our production team.

Kind regards, 

on behalf of

Dr. Heather Macdonald 

Academic Editor

PLOS ONE